# The Impact of Oxygen Pulse and Its Curve Patterns on Male Patients with Heart Failure, Chronic Obstructive Pulmonary Disease, and Healthy Controls—Ejection Fractions, Related Factors and Outcomes

**DOI:** 10.3390/jpm12050703

**Published:** 2022-04-28

**Authors:** Ming-Lung Chuang, Chin-Feng Tsai, Kwo-Chang Ueng, Jui-Hung Weng, Ming-Fong Tsai, Chien-Hsien Lo, Gang-Bin Chen, Sung-Kien Sia, Yao-Tsung Chuang, Tzu-Chin Wu, Pan-Fu Kao, Meng-Jer Hsieh

**Affiliations:** 1Division of Pulmonary Medicine and Department of Internal Medicine, Chung Shan Medical University Hospital, Taichung 40201, Taiwan; cshy276@csh.org.tw (G.-B.C.); cshy041@csh.org.tw (T.-C.W.); 2School of Medicine, Chung Shan Medical University, Taichung 40201, Taiwan; cshy230@csh.org.tw (C.-F.T.); cshy412@csh.org.tw (K.-C.U.); cshy695@csh.org.tw (J.-H.W.); cshy562@csh.org.tw (S.-K.S.); cshy638@csh.org.tw (Y.-T.C.); cshy1219@csh.org.tw (P.-F.K.); 3Division of Cardiology, Department of Internal Medicine, Chung Shan Medical University Hospital, Taichung 40201, Taiwan; cshy1420@csh.org.tw; 4Department of Nuclear Medicine, Chung Shan Medical University Hospital, Taichung 40201, Taiwan; 5Department of Nuclear Medicine, Chiayi Chang Gung Memorial Hospital, Chang Gung Medical Foundation, Chiayi 61301, Taiwan; tsaimifo@cgmh.org.tw; 6Institute of Medicine, School of Medicine, Chung Shan Medical University, Taichung 40201, Taiwan; 7Department of Respiratory Therapy, Chang Gung University, Taoyuan 33303, Taiwan; 8Department of Pulmonary and Critical Care Medicine, Chiayi Chang Gung Memorial Hospital, Chang Gung Medical Foundation, Chiayi 61301, Taiwan

**Keywords:** heart failure with reduced or mildly reduced ejection fraction, chronic obstructive pulmonary disease, ejection fraction, exercise testing

## Abstract

Oxygen pulse (O_2_P) is a function of stroke volume and cellular oxygen extraction and O_2_P curve pattern (O_2_PCP) can provide continuous measurements of O_2_P. However, measurements of these two components are difficult during incremental maximum exercise. As cardiac function is evaluated using ejection fraction (EF) according to the guidelines and EF can be obtained using first-pass radionuclide ventriculography, the aim of this study was to investigate associations of O_2_P%predicted and O_2_PCP with EF in patients with heart failure with reduced or mildly reduced ejection fraction (HFrEF/HFmrEF) and chronic obstructive pulmonary disease (COPD), and also in normal controls. This was a prospective observational cross-sectional study. Correlations of resting left ventricular EF, dynamic right and left ventricular EFs and outcomes with O_2_P% and O_2_PCP across the three participant groups were analyzed. A total of 237 male subjects were screened and 90 were enrolled (27 with HFrEF/HFmrEF, 30 with COPD and 33 normal controls). O_2_P% and the proportions of the three types of O_2_PCP were similar across the three groups. O_2_P% reflected dynamic right and left ventricular EFs in the control and HFrEF/HFmrEF groups, but did not reflect resting left ventricular EF in all participants. O_2_PCP did not reflect resting or dynamic ventricular EFs in any of the subjects. A decrease in O_2_PCP was significantly related to nonfatal cardiac events in the HFrEF/HFmrEF group (log rank test, *p* = 0.01), whereas O_2_P% and O_2_PCP did not predict severe acute exacerbations of COPD. The findings of this study may clarify the utility of O_2_P and O_2_PCP, and may contribute to the currently used interpretation algorithm and the strategy for managing patients, especially those with HFrEF/HFmrEF. (Trial registration number NCT05189301.)

## 1. Introduction

Oxygen pulse (O_2_P) is a function of stroke volume and cellular oxygen extraction [1] and may reflect circulatory stroke work (O_2_P × systolic blood pressure) [2]. Hence, peak O_2_P is used extensively in cardiology as a marker of maximal V’O_2_ and survival [3,4,5,6]. A low peak O_2_P usually indicates low stroke volume caused by primary heart disease based on the assumption that oxygen extraction by muscle cells increases linearly when exercising [1] and the exclusion of coexisting anemia, carboxyhemoglobinemia, arterial oxyhemoglobin desaturation or right-to-left shunt, non-heart-related diseases (such as pain, musculoskeletal disease, ventilatory insufficiency) [1] and/or low arterial-venous oxygen content difference [1,7].

In patients with chronic obstructive pulmonary disease (COPD), although peak O_2_P% ≤ 80%, usually normalized as %predicted [1], has been shown to indicate low stroke volume [8] but not necessarily due to primary heart disease [7,8,9]. Rather, it can be caused by low exercise intensity due to ventilatory limitation [1] or thoracic issues [8,10,11,12,13] and even depleted fat-free mass index [14].

However, stroke volume and oxygen extraction by muscle cells have never been directly measured during incremental maximum exercise because the measurements need invasive and sophisticated methods. As left ventricular ejection fraction (LVEF) is still the gold standard for classifying the type of heart failure and prognosis [15,16], and measurement of ejection fraction during peak exercise can be fulfilled with first-pass radionuclide ventriculography, the relationships between ejection fraction during peak exercise and O_2_P% can be investigated.

In addition, O_2_P curve pattern (O_2_PCP) is the trajectory of O_2_P during exercise, and therefore represents continuous monitoring of O_2_P. O_2_PCP is different in patients with obstructive airway disease and heart disease [7]. However, three types of O_2_PCP have been reported in patients with dyspnea [17] and COPD: plateau (P), decreasing (D), and increasing (I) types [18]. Furthermore, the P and D types are signs of cardiac dysfunction or myocardial ischemia in subjects with various heart diseases [17,19,20], and their presence should prompt additional cardiovascular work-up [19,20]. However, none of the O_2_PCP types are related to an ischemic heart in patients with COPD [18], and the relationships between O_2_PCP and ejection fraction in normal controls and in patients with COPD and chronic heart failure are unclear.

The aims of this study were to investigate the following: (1) differences in O_2_P% and the proportions of the O_2_PCP types across the three groups (normal subjects, patients with COPD, and patients with heart failure with reduced or mildly reduced ejection fraction (HFrEF/HFmrEF)), (2) variables related to O_2_P% and O_2_PCP including resting and dynamic ventricular ejection fractions, and (3) factors related to cardiac events and acute exacerbations of COPD [21]. The findings of this study may clarify the utility of O_2_P and O_2_PCP in different populations and thus may critically affect the currently used interpretation algorithm and strategy for managing patients with HFrEF/HFmrEF and COPD.

## 2. Materials and Methods

### 2.1. Study Design

This was a prospective observational cross-sectional study on O_2_P% and O_2_PCP across three groups (CHF, COPD, and normal controls) and to investigate the clinical implications of O_2_P and O_2_PCP. The Institutional Review Board of Chung Shan Medical University Hospital (CS16174 and CS19014) and Chang-Gung Memorial Hospital (201700899A3) approved this study and all participants provided written informed consent. The study was conducted in compliance with the Declaration of Helsinki. (Trial registration number NCT05189301.)

### 2.2. Subjects

We enrolled healthy subjects, participants with moderate to severe COPD alone, and participants with symptomatic HFrEF/HFmrEF alone from two institutions in Taiwan, from 1 August 2017 to 8 July 2020, and followed them until 27 May 2021. The inclusion criteria were: (1) age 40–80 years; (2) body mass index 18–30 kg/m^2^; (3) male sex. Only male subjects were enrolled as few female participants had COPD in Taiwan [22]. All of the participants had been in a stable clinical condition for at least 1 month before participating in the study. Participants with diabetes mellitus, uncontrolled hypertension, arrhythmia, cancer, liver, renal or autoimmune diseases, or anemia were excluded. However, participants with well-controlled diabetes mellitus were included in the HFrEF/HFmrEF group as these conditions often co-exist. Physical activity was not encouraged or limited during the study period. None of the participants had contraindications for cardiopulmonary exercise tests (CPETs).

*COPD*. Participants with COPD were referred by pulmonologists, and they all had respiratory symptoms, risk factors and a post-bronchodilator forced expired volume in one second/forced vital capacity of <0.7 in accordance with the Global Initiative for Chronic Lung Disease criteria [23]. Participants with a significant post-bronchodilator effect (increase in forced expired volume in one second >12% and 200 mL from baseline) were excluded. Participants combined with HFrEF/HFmrEF were also excluded.

*HFrEF/HFmrEF*. Participants with HFrEF/HFmrEF who had reduced and mildly reduced ejection fractions were referred by cardiologists, and they were enrolled if they had New York Heart Association functional class (NYHA) I-III, and risk factors. A left ventricular ejection fraction using two-dimensional echocardiography (2DLVEF) <45–50% was obtained within 2 months before or after commencing the study [15]. Participants combined with lung diseases were also excluded.

*Normal controls*. Healthy subjects were recruited among the hospital staff and the local community through personal contacts. They were free of known significant diseases.

### 2.3. Measurements

*Functional capability assessment.* The daily functional activities of the subjects were assessed using the oxygen-cost diagram (OCD), NYHA scale and modified Medical Research Council (mMRC) scale [24]. They were asked to indicate a point on an OCD, a 100 mm-long vertical line with everyday activities listed alongside the line, above which breathlessness limited them. The distance from zero was measured and scored in cm. The COPD Assessment Test (CAT) questionnaire was used to evaluate the quality of life in patients with COPD. For simplicity, all of the participants completed all functional capability assessments in this study [25].

*Blood test.* High-sensitivity C-reactive protein (hs-CRP) and N terminal pro-brain natriuretic peptide (NT-proBNP) were measured using near infrared particle immunoassay rate method and fluorescence assay, respectively, in all participants. Lipid profile was measured using enzymatic method in the HFrEF/HFmrEF group. The laboratory has been qualified by Taiwan Accreditation Foundation ISO15189 and the site number is 1817.

*Two-dimensional echocardiography.* Two-dimensional echocardiography (EPIQ7C, Philips, Seattle, WA, USA) was performed in all participants with COPD and HFrEF/HFmrEF, and in randomly selected controls. Two-dimensional echocardiography was performed by two experienced technicians and reviewed by cardiologists who were blinded to all clinical data. Parasternal, apical and subcostal studies were conducted [15].

*Pulmonary function tests.* All pulmonary medications were discontinued at least 24 h before pulmonary function studies. Spirometry, lung volumes and diffusing capacity (D_L_CO) were measured with a body plethysmograph (MasterScreen Body, Germany) before bronchodilator tests. Post bronchodilator data of spirometry were obtained after inhaling 400 μg of fenoterol HCl in the COPD group. We have previously reported the predicted values currently used at our institute. The predicted values are in line with our previous report [26].

*CPET.* Pulmonary gas exchange was measured in each subject on different days within 1 month after lung function tests. Short-acting and long-acting beta bronchodilators/muscarinic antagonists (LABA/LAMA) were withheld 4–6 h and ≥12 h before the test, respectively. Antianginals, β-blockers and calcium antagonists were stopped 24 h before each test. Gas exchange equipment including a face mask connected to a turbine pneumotachograph was used to measured V’O_2_ (mL/min, STPD), CO_2_ output (V’CO_2_) (mL/min), and minute ventilation (V’_E_) (L/min, BTPS) breath by breath (MasterScreen CPX™, Germany), and then the data were averaged and reported at 15 s intervals of each stage. We used well-established references for predicted normal values of peak V’O_2_ and O_2_P [1,27]. Oscillatory breathing was defined according to Leite’s study [28]. Anaerobic threshold (AT) was measured using modified V-slope and nadir of V’_E_/V’O_2_ methods [7]. For each test, 12-lead electrocardiograms (EKG), arterial oxyhemoglobin saturation, and blood pressure were recorded. In EKG, horizontal or down-sloping ST segment depression ≥1 mm or up-sloping ≥1.5 mm at 80 ms after the J point was considered to indicate ischemic change [29]. An electromagnetically braked cycle ergometer (Lode, The Netherlands) was used. The exercise test protocol was a 3 min period of rest followed by 3 min period of unloaded exercise, followed by ramp-pattern-loaded exercise with a work rate per stage selected according to the OCD scores [30].

*Development of smoothing techniques for O_2_P curve.* All O_2_P data from resting to peak exercise were smoothed using computer software (Microcal Origin v 4.1, OriginLab Corp, Northampton, MA, USA) [18]. Increases <0.5 mL/beat or progressive decreases in O_2_ pulse that persisted up to exercise cessation for at least 1 min were defined as a P or a D trajectory, respectively [20]. This algorithm was used to determine O_2_PCP, and the intra- and inter-rater agreements were 0.86 and 0.69, respectively, using κ statistics. The final types of O_2_PCP were decided by consensus of two investigators (M.L.C. and M.J.H.) before analyzing the other study data.

First-pass radionuclide ventriculography (FPRV) and gated myocardial perfusion single-photon emission computed tomography (SPECT). Radionuclide assessment of ventricular performance was performed (Symbia T2, Germany) [31]. A right ventricle ejection fraction (FPRVEF) ≥45% and LVEF (FPLVEF) ≥55% were considered to be normal [32].

FPRV was performed in all participants with COPD and HFrEF/HFmrEF and in randomly selected controls, due to ethical considerations. After reaching peak exercise, the subjects immediately lay down on a pallet and a bolus of 10 mCi of high specific activity technetium-99m sestamibi was injected via a right antecubital vein or external jugular vein [32]. Another bolus of 25–30 mCi of technetium-99m sestamibi was injected for resting images approximately 1 h after exercise to determine myocardial wall motion abnormalities and ischemia using SPECT. The severity of myocardial ischemia was defined according to summed stress score (SSS), summed rest score (SRS), and summed difference score (SDS = the difference between SSS and SRS) [33]. A post-stress LVEF > 50% with SSS 4–13 indicated a low risk of a cardiac event or death, and a post-stress LVEF < 30% indicated a high risk [29].

*Outcomes.* The participants were followed from the day of CPET to the first pulmonary or cardiac event, date of censor if they were lost to follow-up, or the end of the study (no-event censored cases), whichever occurred first. The outcomes of the healthy controls were evaluated using scripted telephone interviews. In the participants with COPD and HFrEF/HFmrEF, follow-up consisted of reviewing hospital medical records. Pulmonary events were defined as pulmonary death or severe acute exacerbation of COPD (SAECOPD), defined as an acute worsening of respiratory symptoms requiring antibiotics and/or increasing dose of corticosteroids during hospitalization [34]. The SAECOPD rate was calculated as per person per year (PPPY) [35]. Cardiac events were defined as either cardiac death or pulmonary edema due to HFrEF/HFmrEF for which hospitalization was required, or nonfatal myocardial infarction for which percutaneous coronary revascularization was performed. Risk factors of pulmonary and cardiac events were analyzed using a Cox regression model, and Kaplan–Meier survival curves with log rank analysis were used for survival/event analysis.

### 2.4. Statistical Analysis

The sample size was estimated to be around 30 to achieve 80% power to detect a difference of 0.5 between the null hypothesis correlation of 0 and alternative hypothesis correlation of 0.5 using a two-sided hypothesis test with a significance level of 0.05. The calculation was based on the following formula: *n* = [(Z_α_ + Z_β_)/w]^2^ where w is Fisher’s z transformation of the sample correlation coefficient r: w = 0.5 × ln[(1 + r)/(1 − r)] [36]. For each measured variable, the comparisons were planned a priori. Data were summarized as mean ± standard deviation or median (25th–75th percentiles) when appropriate, and therefore p values were calculated using one-way analysis of variance (ANOVA) or Kruskal–Wallis one-way ANOVA for group comparisons when appropriate. The chi-square test or Fisher’s exact test for multiple comparisons was used in contingency table analysis for categorical variables. Pearson’s correlation coefficients were further used for quantifying pairwise relationships among the variables of interest. To assess associations between the variables of interest and pulmonary or cardiac events during follow-up, Kaplan–Meier survival curves were constructed, and the log-rank test was used. Receiver operating characteristic curves were used for the variables of interest to determine their optimal prognostic threshold values for pulmonary and cardiac events. Hazard ratios (HRs) were calculated and tested using a Cox regression model treating each variable of interest as continuous. All statistical analyses were performed using NCSS statistical software (NCSS 9, NCSS, LLC., Kaysville, UT, USA). Statistical significance was set at a two-sided *p ≤* 0.05.

## 3. Results

A total of 237 male subjects were screened, of whom 147 were excluded with the reasons reported in Figure 1. The remaining 90 subjects were enrolled, including 30 with COPD, 27 with HFrEF/HFmrEF and 33 normal controls (Table 1). There was no significant difference in V’O_2peak_% between the COPD and HFrEF/HFmrEF groups; however, the COPD group had a higher hs-CRP level, worse lung function and gas exchange at anaerobic threshold, and more dyspneic and no oscillatory breathing during exercise (Table 1 and Table 2). In comparison, the HFrEF/HFmrEF group as expected had the higher NT-proBNP level and lower resting and dynamic ejection fractions (2DLVEF 41.6 ± 6.6%, FPLVEF 42.0 ± 11.7%) and pulse pressure (71.7 ± 28.7 mmHg) at peak exercise, and five patients had oscillatory breathing (19.2%).

*O_2_P% versus O_2_PCPs.* Peak O_2_P% strongly reflected peak aerobic function and was not different or related to resting ejection fraction in any of the subjects (Table 2 and Table 3, all *p* = NS). In addition, peak O_2_P% reflected dynamic RVEF and LVEF to different extents in the control and HFrEF/HFmrEF groups, but not in the COPD group. This may have been because peak O_2_P% and dynamic ejection fractions were affected to different extents by dynamic hyperinflation in the COPD group (|r| = 0.53 versus 0.30–0.33, *p* = 0.003 versus ≤0.1). Peak O_2_P% did not predict complications in either the COPD or HFrEF/HFmrEF group.

Of the three types of O_2_PCP (Figure 2), the D type was rare, whereas the I and P types were common, with similar incidence rates across the three groups (Table 2). O_2_PCPs were not related to resting or dynamic ejection fractions in any of the participants (Table 4). In the HFrEF/HFmrEF group, D-O_2_PCP was a marker of high NT-proBNP, poor quality of life and exercise physiology (including peak O_2_P%), and predicted nonfatal cardiac events (positive predictive value of 75%).

*Factors related to outcomes.* There were no cardiac deaths in the HFrEF/HFmrEF group; however, six episodes of nonfatal myocardial infarction and HFrEF/HFmrEF-related pulmonary edema were recorded during 5–1129 days of follow-up. Cox regression analysis revealed that the risk factors related to cardiac events were the number of diseased coronary arteries, CAT score, total cholesterol, and D-O_2_PCP, but not diabetes mellitus, hypertension, V’O_2peak_/kg < 16.5 mL/min/kg, V’_E_/V’CO_2nadir_ > 34.6, oscillatory breathing, resting LVEF <30%, radioisotope images for LVEF or myocardial ischemia (Figure 3). The log rank test revealed that D-O_2_PCP (Figure 4, *p* = 0.012), CAT ≥ 5 (*p* = 0.01), and ≥2 diseased coronary arteries (*p* = 0.05) were related to nonfatal cardiac events.

During 12–1370 days of follow-up, there were no pulmonary deaths; however, seven SAECOPD episodes were recorded (0.21 ± 0.65 PPPY) in the COPD group. Cox regression analysis revealed that the risk factors for SAECOPD were SAECOPD in the previous 12 months, age, and poor functional capability; the beneficial factors were exercise capability and D_L_CO% (Figure 3, all *p* < 0.05). One episode of nonfatal myocardial infarction occurred in one participant with I-O_2_PCP 164 days after CPET even though SPECT showed a low risk of cardiac events. The log rank test revealed that SAECOPD in the previous 12 months (*p* = 0.0001) and impaired functional capability were related to SAECOPD (Figure 5).

## 4. Discussion

The main findings of this study are that there were no differences in the incidence of the three types of O_2_PCP or peak O_2_P% across the three groups of subjects. In the normal controls, the types of O_2_PCP were not associated with the clinical characteristics or physiology, whereas peak O_2_P% reflected dynamic cardiac function to some extent. In the COPD group, neither O_2_PCP nor peak O_2_P% was related to cardiac function or SAECOPD. In the HFrEF/HFmrEF group, D-O_2_PCP was related to cardiac events, and peak O_2_P% was related to dynamic cardiac function.

*Associations of peak O_2_P and O_2_PCP with cardiac function.* At peak exercise, muscle O_2_ extraction cannot be assumed to be constant because a substantial proportion of the general population has mitochondrial myopathy [20,37], and substantial variations in arterial-venous oxygen content gradient have been reported in normal subjects [38]. Accordingly, peak O_2_P may not precisely reflect stroke volume. However, although peak O_2_P% was related to dynamic ejection fractions in the control and HFrEF/HFmrEF groups, the relationship was not significant in the COPD group. This may be because peak O_2_P% and dynamic ejection fractions are affected by dynamic hyperinflation to different extents in COPD. Although it is unclear whether or not peak O_2_P% represents stroke volume, it may have represented dynamic ejection fractions in the control and HFrEF/HFmrEF groups, in which ejection fraction is still the gold standard to classify the type of heart failure and prognosis [15,16]. It is unclear why peak O_2_P% and O_2_PCP in the disease groups were similar to the controls in this study. It is possible that the severity of disease was mild in some patients, and that O_2_PCP and O_2_P% are related to other factors in addition to LVEF and the Global Initiative for Chronic Lung Disease stage. V’O_2_% and heart rate % were indeed different at peak exercise across the three group (Table 2), with the highest value in the controls. When V’O_2_ was divided by heart rate (i.e., O_2_P) at peak exercise, the difference in O_2_P% was insignificant across the three groups. Nevertheless, we speculate that the contribution of muscle oxygen extraction to O_2_P may have compensated stroke volume in the HFrEF/HFmrEF group. This notion warrants further investigations.

P-O_2_PCP has been reported to be superior to peak O_2_P% in diagnosing cardiac causes of exercise limitation [17,19,33,39]. Belardinelli et al. reported that when adding flattened O_2_PCP duration and decreased second segment slope of ΔV’O_2_/Δwork rate (2nd slope) to EKG stress testing, the diagnostic rate was much improved [33]. However, O_2_PCP appears to be nonspecific because P-O_2_PCP was noted in 1/3 of the healthy controls and P- and D-O_2_PCPs were noted in approximately 60% of the COPD and HFrEF/HFmrEF groups in this study, indicating that muscle oxygen extraction must have plateaued after AT because stroke volume normally plateaus after AT. O_2_PCP was not related to resting EF, which is consistent with a previous study [18], or to dynamic ejection fractions in this study. Nevertheless, patients with mitochondrial myopathy have been reported to have the I-type of O_2_PCP [20]. In this context, stroke volume must be supernormal when tissue oxygen extraction plateaus, otherwise tissue oxygen extraction would increase when stroke volume normally plateaus. Several types of mitochondrial disorders have been associated with normal peak cardiac output with decreased tissue oxygen extraction, but no heart rate or stroke volume data have been reported [40]. Taken together, O_2_PCP and O_2_P% are complex and reflect aerobic function (r = 0.81–0.91), and peak O_2_P reflected dynamic cardiac function to different extents in the normal controls and HFrEF/HFmrEF group.

In the COPD group in this study, although peak O_2_P and O_2_PCP did not reflect cardiac function, they were related to functional and aerobic capabilities and dynamic hyperinflation. Dynamic hyperinflation strongly affected peak O_2_P% but only marginally affected dynamic ejection fractions (Table 4). In this context, the relationship between peak O_2_P% and dynamic ejection fractions was inconsistent. D-O_2_PCP has been related to motivation to perform exercise [18], and hence the subjects may have had higher self-assessed exercise capability.

*Outcomes.* In the present study, D-O_2_PCP, CAT score ≥5, and the number of diseased coronary arteries were related to nonfatal cardiac events (Figure 3 and Figure 4, log rank test *p* = 0.05–0.01). To the best of our knowledge, this is the first report on the relationship between D-O_2_PCP or CAT score ≥5 and nonfatal cardiac events in patients with HFrEF/HFmrEF.

In total, 6 of 27 (22.2%) subjects with HFrEF/HFmrEF had nonfatal cardiac events, and 7 of 30 (23.3%) subjects with COPD had SAECOPD. It could be argued that these cardiovascular and pulmonary event rates were very low. However, previous studies have reported a 1-year mortality rate of 25% in patients with advanced HF [21], and 0.25 SAECOPD events per year [41]. In comparison, there were five to six events in our HFrEF/HFmrEF group and seven to eight events in our COPD group.

CAT and mMRC scores have been widely used to assess the health status of patients with COPD [23,25,42]. There was no significant difference in the total CAT score between the COPD and HFrEF/HFmrEF groups in the present study, suggesting that the subjects with HFrEF/HFmrEF had similar symptoms to those with COPD, and this may have interfered with their quality of life (Table 1). In receiver operating characteristic analysis in the HFrEF/HFmrEF group, a CAT score ≥5 had the best diagnostic ability to identify cardiac events (AUC = 0.80, *p* = 0.0007) and was also related to cardiac events on follow-up (log rank test, *p* = 0.01). In comparison, the subjects with CAT ≥5 had more cough, sputum, and chest tightness and less energy and higher NYHA score (1.3 ± 0.4 versus 1.9 ± 0.5), lower V’O_2peak_% (78.3 ± 17.8% versus 64.3 ± 13.8%) and O_2_P% (93.8 ± 17.7% versus 79.4 ± 17.1%, all *p* < 0.05).

In the literature, age, diabetes, hypertension, the length of the diseased coronary arteries, bare metal stenting and initial minimal luminal diameter < 3 mm [42], P-O_2_PCP and the second segment slope of ΔV’O_2_/Δwork rate [34], oscillatory breathing [43], resting FPRVEF < 20% [44], V’O_2peak_ < 16.5 mL/min/kg plus V’_E_/V’CO_2_ > 34.6 [45], and LVEF% (>50% or <30%) plus the amount of myocardial ischemia (large amount or none) [29] have been related to cardiac events; however, none were related to nonfatal cardiac events in this study (all HRs, p = NS). It is possible that these cutoff values were set too high. A recent study reported that changes in therapy for HFrEF/HFmrEF altered the sensitivity and specificity of CPET variables for the prognosis of HFrEF/HFmrEF [46].

Many risk factors for a first SAECOPD and subsequent SAECOPD have been reported [35,41]. These include a prior SAECOPD, severe airflow limitation, poor health status, old age, radiologic evidence of emphysema, and high WBC count. In this study, D_L_CO%, functional capability such as CAT, mMRC, and NYHA scores, and many aerobic function parameters were related to SAECOPD. Recently, D_L_CO% < 50% has also been reported to be as good as or even a better prognostic marker than forced expired volume in one second% < 50% for SAECOPD [47]. Aerobic function has been reported to be related to survival, but not yet to SAECOPD [48]. By linking to pulmonary rehabilitation, it is possible that aerobic function reduces SAECOPD [49].

## 5. Study Limitations

In this study, we used peak O_2_P% as suggested in the literature [1] rather than absolute peak exercise O_2_P because O_2_P can be affected by anthropometrics. This is supported by reports that absolute peak exercise O_2_P did not add any prognostic information to V’O_2__peak_ in patients with HFrEF/HFmrEF [50], but that peak O_2_P adjusted for body weight in kg did add such information [46]. However, it is unclear whether to use peak exercise O_2_P% or that adjusted by kg. It may be argued that the severity of HFrEF/HFmrEF in the patients in this study was mild, and that including more severe patients may have resulted in different findings. However, we enrolled heart failure patients with reduced and mildly reduced ejection fractions, of whom approximately 1/3 had the I type of O_2_PCP and a V’O_2peak_ > 80%predicted, significantly higher than that of the subjects with the P and D types (Table 4). The results indicate that CPET provides additional information to 2DLVEF for patients with HFrEF/HFmrEF. Alternatively, stroke volume/end-systolic volume measured with magnetic resonance imaging and echocardiography, tricuspid annular plane systolic excursion, and RV myocardial performance index are potential markers of heart function. Increases of at least 5% in FPRVEF and FPLVEF are other markers of a normal ventricular response to maximal graded exercise. Including them may have made the findings of RV and LV function more convincing. It may be argued that without serial measurements of ejection fraction during exercise, there really is no way of knowing whether O_2_PCP types do in fact represent ejection fraction types. However, we could not clarify this issue due to the technical limitations of FPRV. First, it is not possible to repeat injections of isotopes at every stage during exercise. Second, it has to be finished within seconds per test. Thus, FPRV is usually performed at peak exercise or during steady state of a constant work rate exercise. We excluded COPD patients with heart disease due to the study design and aims, and thus the results can only be applied to the inclusion criteria of the current study. Hence, using the recommendations of this study in patients with both COPD and coronary artery disease should be performed with caution. In addition, emphysema score was not measured with high-resolution computed tomography, so we could not evaluate whether O_2_PCPs and SAECOPD were affected by emphysema. As the number of cases in this study was small, it was difficult to conduct the log rank test with two variables of interest combined. To confirm this notion, further studies with a larger number of cases are warranted. As the duration of follow-up in this study was short, further studies are warranted. Lastly, female subjects were excluded from this study, and further studies are warranted to investigate any sex-related differences. A summary of the limitations is as follows: (1) O_2_P adjusted for body weight in kg or predicted value warrants further studies; (2) further studies to validate O_2_P%and O_2_PCP using techniques other than 2DLVEF and FPRVEF are warranted; (3) using the recommendations of this study in patients who meet the exclusion criteria should be performed with caution; (4) further studies with a larger number of cases, a longer duration of follow-up, and including female subjects and more severe patients with HFrEF/HFmrEF are warranted.

## 6. Conclusions

By evaluating O_2_PCP and O_2_P% during a ramp-pattern exercise test with gas exchange and FPRV and SPECT, this study reports the different relationships between ejection fraction and O_2_PCPs and O_2_P% in healthy subjects and those with COPD and HFrEF/HFmrEF, and the prognosis of patients with HFrEF/HFmrEF. However, as the number of cases in this study was small, further studies are warranted to confirm this issue.

## Figures and Tables

**Figure 1 jpm-12-00703-f001:**
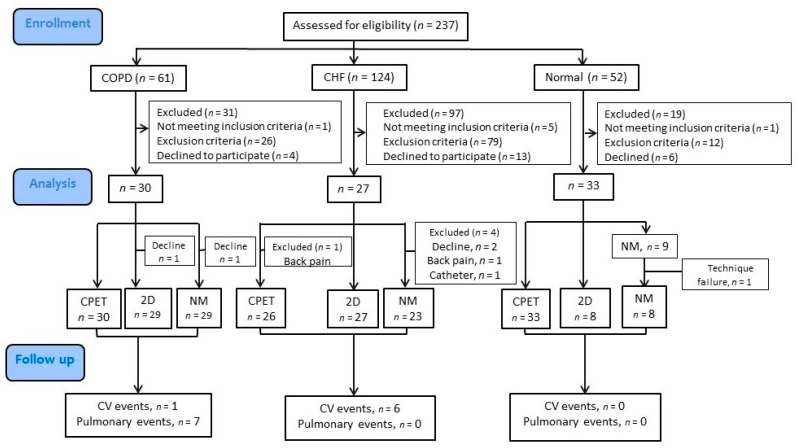
Flow chart. A total of 237 subjects were assessed for eligibility. Thirty subjects had chronic obstructive pulmonary disease (COPD), 27 had heart failure with reduced or mildly impaired ejection fraction (HFrEF/HFmrEF), and 33 healthy subjects were enrolled. Participants with HFrEF/HFmrEF were enrolled if they had New York Heart Association functional class (NYHA) I-III and relevant risk factors. A left ventricular ejection fraction using two-dimensional echocardiography (2DLVEF) <45–50% was obtained within 2 months before or after commencing the study. All of the participants with COPD had respiratory symptoms, risk factors and a post-bronchodilator forced expired volume in one second (FEV_1_)/forced vital capacity (FVC) of <0.7 without a significant post-bronchodilator effect. Healthy subjects were recruited among the hospital staff and the local community through personal contacts. They were free of known significant diseases. A total of 147 subjects were excluded due to the reasons shown. Participants with diabetes mellitus, uncontrolled hypertension, arrhythmia, cancer, liver, renal or autoimmune diseases, or anemia were excluded. However, participants with well controlled diabetes mellitus were included in the HFrEF/HFmrEF group as these conditions often co-exist. For details about the inclusion and exclusion criteria of the participants, please refer to the text. CPET: cardiopulmonary exercise testing; 2D: 2-dimensional echocardiography; NM: nuclear medicine for 1st pass right ventriculography. Cardiac events did not include cerebrovascular accidents.

**Figure 2 jpm-12-00703-f002:**
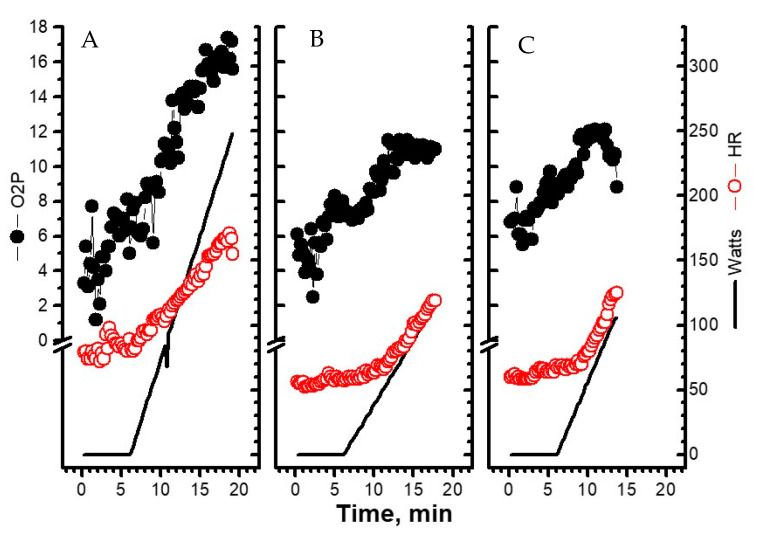
Oxygen pulse curve patterns. (**A**) A representative image of the increasing pattern. (**B**) A representative image of the plateau pattern. (**C**) A representative image of the decreasing pattern. HR: heart rate, watts: workload, O_2_P: oxygen pulse.

**Figure 3 jpm-12-00703-f003:**
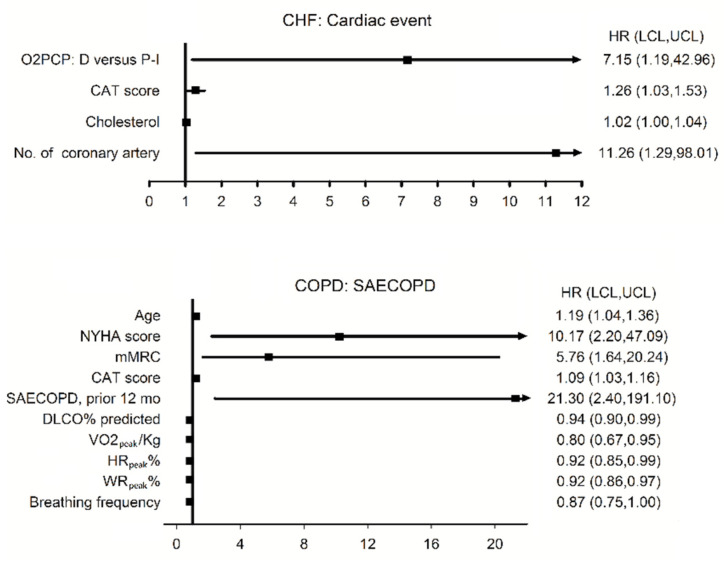
Forest plots of risk factors for non-fatal cardiac events in the chronic heart failure (CHF) group (upper panel, *n* = 27) and for severe acute exacerbations of chronic obstructive pulmonary disease (SAECOPD) in the COPD group (lower panel, *n* = 30) using Cox regression analysis and the range of LCL and UCL ≥1 or ≤1 indicates significance. HR: hazard ratio; O_2_PCP: oxygen pulse curve pattern; D versus P-I: type decreasing versus types plateau and increasing; CAT: COPD assessment test; no. of coronary artery: number of diseased coronary artery; NYHA: New York Heart Association; mMRC: modified medical research council; D_L_CO: diffusing capacity of lung for carbon monoxide; V’O_2peak_: oxygen uptake at peak exercise; HR_peak_: heart rate at peak exercise; WR_peak_: work rate at peak exercise; breathing frequency: breathing frequency at peak exercise.

**Figure 4 jpm-12-00703-f004:**
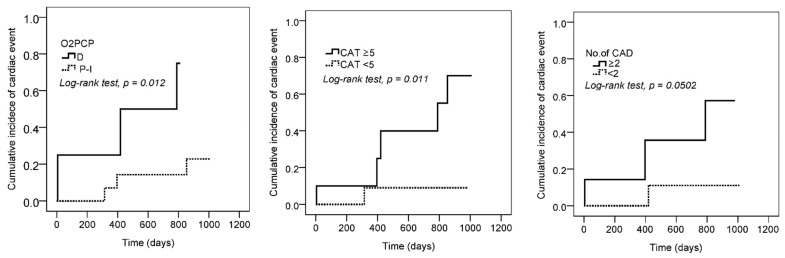
Kaplan–Meier survival curves of cardiac events are constructed and the log-rank test is used according to the following: O2PCP: oxygen pulse curve patterns (solid line indicates the decreasing pattern and dashed line indicates the increasing and plateau patterns, log rank, *p* = 0.012); CAT: COPD assessment test (*p* = 0.01); No. of CAD: the number of diseased coronary artery (*p* = 0.05) were related to nonfatal cardiac events.

**Figure 5 jpm-12-00703-f005:**
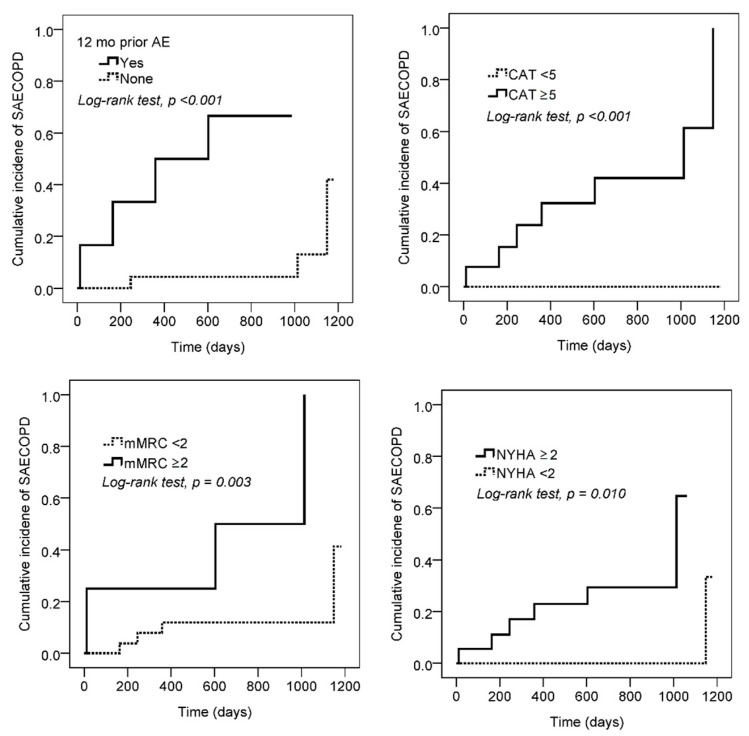
Kaplan–Meier survival curves of severe acute exacerbation of chronic obstructive pulmonary disease (SAECOPD) are constructed and the log-rank test is used according to SAECOPD in the previous 12 months (*p* = 0.0001), modified Medical Research Council score (mMRC) (*p* = 0.003), NYHA (*p* = 0.01), and COPD assessment test score (CAT) (*p* = 0.03).

**Table 1 jpm-12-00703-t001:** Demographic data, symptom scores, blood tests, physiological data at rest and medication use.

Group	CHF		COPD		Controls		ANOVA
	mean	SD	mean	SD	mean	SD	*p*-value
*n*	27		30		33		
Demographics and history							
Age	57.7 ^‡^	9.2	**68.3**	7.5	61.8	9.0	^‡^
Height, cm	167.4	4.2	166.0	4.9	166.7	5.2	^NS^
Body mass index, kg/m^2^	26.6 ^‡^	3.2	**23.1**	3.2	24.8	2.7	^†^
Smoke, pack-year	28.9 ^‡^	28.1	**67.9**	38.8	2.9	9.7	^‡^
SAECOPD 12 mo prior, count	^NA^		6		^NA^		^NA^
Cardiac event/SAECOPD on follow-up, count	6		7		^NA^		^NA^
SAECOPD on follow-up, rate, PPPY	^NA^		0.21	0.65	^NA^		^NA^
Functional capability and quality of life							
NYHAfc I, II, III, IV, *n*	14/12/1/0		12/13/5/0		**32** **^∨^/0/0/0**		^‡^
Borg dyspnea score @ rest, 0/0.5/1/2, *n*	13/8/6/0		18/9/1/2		**32/0/1/0**		^†^
mMRC 0–4, *n*	17/7/3/0/0		12/14/4/0/0		**33/0/0/0/0**		^‡^
Oxygen-cost diagram, cm	7.5	0.9	7.0	1.6	**8.3**	1.0	^†^
CAT, summed score	3.8	3.7	6.0	6.9	**0.5**	1.0	^‡^
Blood test							
hs-CRP, mg/dL	0.2 *	0.3	**2.0**	4.0	0.2	0.2	**
NT-proBNP, pg/mL	**358.3** ^‡^	295.8	46.0	36.0	45.3	36.2	^‡^
Hemoglobin, gm/dL	15.3	1.2	14.8	1.6	14.7	1.1	^NS^
Cholesterol, mg/dL	163.0	45.1	^NA^		^NA^		^NA^
2DLVEF							
resting, %	**41.6** ^‡^	6.6	65.5	8.8	61.3 ^	5.9	^‡^

CHF: congestive heart failure; COPD: chronic obstructive pulmonary disease (stage 1/2/3/4, *n* = 2/22/5/1); ^ *n* = 8 in randomly selected subjects. ^∨^ missing one data. CHF: 23 ischemic cardiomyopathy and 4 dilated cardiomyopathy; SAECOPD: severe acute exacerbation of COPD; PPPY: per person per year; mMRC: modified medical research council; NYHAfc: New York Heart Association functional classification; CAT: COPD assessment test; hs-CRP: high-sensitivity C-reactive protein; NT-proBNP: N terminal pro-brain natriuretic peptide; 2DLVEF: left ventricular ejection fraction measured with two-dimensional echocardiography. Bolded digits indicated the highest or lowest values in comparison of variables of interest across the three groups. Symbols in the CHF group column indicate comparisons between the COPD and CHF groups. * *p* ≤ 0.05, ** *p* < 0.01, ^†^
*p* < 0.001, ^‡^
*p* < 0.0001. ^NA^ not applicable; ^NS^ not significant.

**Table 2 jpm-12-00703-t002:** Lung function and cardiopulmonary exercise test (CPET) and first pass radionuclide ventriculography (FPRV) (selected presentation for brevity).

Group	CHF		COPD		Controls		ANOVA
	mean	SD	mean	SD	mean	SD	*p*-value
*n*	27		30		33		
Lung function							
FEV_1_ %predicted, %	94.9 ^‡^	13.0	**61.0**	15.5	102.7	12.8	^‡^
FEV_1_/FVC, %	81.4 ^‡^	3.9	**57.5**	10.4	79.8	5.7	^‡^
RV %predicted	102.4 ^‡^	18.1	**138.7**	32.8	101.0	17.5	^‡^
RV/TLC %predicted	106.6 ^‡^	14.3	**137.0**	20.7	101.4	11.9	^‡^
D_L_CO %predicted	89.4 *	12.9	**76.5**	20.4	106.2	15.4	^‡^
CPET							
V’_E_/VCO_2_ at anaerobic threshold	33.1 **	5.7	**38.2**	6.7	31.7	3.6	^‡^
at peak exercise							
V’O_2_%pred	72.9	17.7	72.8	18.4	**91.6**	19.3	^‡^
Respiratory exchange ratio	1.06	0.08	1.00	0.09	**1.17**	0.14	^‡^
Heart rate (HR)%predicted	83.0	11.4	82.9	11.6	**95.2**	9.6	^‡^
ΔHR/ΔV’O_2_%, beat/min/L/min	104.0 ^†^	35.2	94.3	21.9	104.1	29.1	^NS^
O_2_P%predicted, %	88.2	18.9	87.4	16.2	96.9	22.5	^NS^
O_2_PCP, type I/P/D, *n* (%)	9/13/4 ^^^ (35/50/15)		14/12/4 (47/40/13)		21/11/1 (64/33/3)		^NS ^^^
ΔV’O_2_/ΔWR, slope 2	9.1	2.6	9.3	2.2	9.3	1.3	^NS^
Blood pressure, systolic, mm Hg	**182.9** *	22.4	203.3	35.2	207.1	25.9	**
Pulse pressure, mm Hg	**71.7** ^‡^	28.7	109.5	32.5	103.2	32.2	^‡^
Breathing frequency, b/min	31.6	7.3	32.8	6.5	**36.4**	10.3	^¶^
ΔBorg dyspnea/ΔV’O_2_	10.5	5.1	**12.7**	7.0	8.8	3.5	*
V_T_/TLC	0.32 ^‡^	0.07	**0.24**	0.06	0.33	0.05	^‡^
SpO_2_, %	96.8 ^‡^	1.5	**92.6**	3.3	96.7	1.2	^‡^
FPRV							
FPRVEF,%	**52.3** ^†^	8.3	63.2	13.0	57.4	12.9	**
FPLVEF,%	**42.0** ^‡^	11.7	63.6	11.7	72.4	3.6	^‡^
SPECT ^^^^^: SSS	25.3 ^‡^	9.9	7.6	2.6	^NA^		^NA^
SDS	0.7	1.6	^NA^		^NA^		^NA^
LVEF post exercise,%	44.8 ^‡^	17.2	70.3	1.7	^NA^		^NA^

CHF: chronic heart failure; COPD: chronic obstructive pulmonary disease; FEV_1_: forced expired volume in one second; FVC: forced vital capacity; RV: residual volume; TLC: total lung capacity; D_L_CO: diffusing capacity of lung for carbon monoxide; SPECT: single-photon emission computed tomography; SSS: summed stress score; SDS: summed difference score, i.e., the difference between SSS and SRS. V’O_2_: O_2_ uptake; V’_E_: minute ventilation; Δ: change; slopes 2: slope 2 between anaerobic threshold and peak exercise using linear regression; O_2_P: oxygen pulse; O_2_PCP: oxygen pulse curve pattern; I: increasing; P: plateau; D: decreasing; FPRVEF: first pass right ventricular ejection fraction; LVEF: left ventricular ejection fraction; V_T_: tidal volume; SpO_2_: oxyhemoglobin saturation measured with pulse oximetry. For FPRV: *n* = 29 for the COPD group, *n* = 23 for the CHF group, and *n* = 8 for the normal group; ^^^: one was excluded due to submaximal exercise caused by back pain; ^^^^: Fisher’s exact test; *p* = 0.14. ^^^^^: *n* = 14 for the COPD group, *n* = 18 for the CHF group. Bolded digits indicated the highest or lowest values in comparison of variables of interest across the three groups. Symbols in the CHF group column indicate comparisons between the COPD and CHF groups. ^¶^ 0.05 < *p* ≤ 0.1, * *p* <0.05 ** *p* < 0.01 ^†^
*p* < 0.001 ^‡^
*p* < 0.0001. ^NA^ not applicable; ^NS^ not significant.

**Table 3 jpm-12-00703-t003:** Relationships between oxygen pulse % predicted and variables of interest in different groups (selected presentation for brevity).

Group, *n*	Normal, 33		COPD, 30		CHF, 27	
	r	*p*-value	r	*p*-value	r	*p*-value
Functional capability/Quality of life						
Oxygen-cost diagram	0.50	0.003		ns	0.37	0.07
Aerobic capability						
V’O_2peak_%	0.91	<0.0001	0.85	<0.0001	0.81	<0.0001
ΔV’O_2_/ΔWR	0.42	0.02	0.60	0.001	0.69	0.0001
Cardiac function						
NT-proBNP		ns		ns	−0.40	0.06
2DLVEF, resting, %		ns		ns		ns
FPLVEF, peak, %		ns	^	ns	0.49	0.02
FPRVEF, dynamic, %	0.74	0.03	^	ns	0.40	0.06
Peak dynamic hyperinflation						
V_T_/TLC		ns	0.53	0.003	0.44	0.03
Lung function						
FEV_1_%		ns	0.51	0.004		ns
D_L_CO%		ns	0.55	0.002		ns
RV/TLC % predicted	−0.45	0.01	−0.34	0.07		ns

V’O_2peak_: oxygen uptake at peak exercise; ΔV’O_2_/ΔWR: slope of oxygen uptake in response to work rate; NT-proBNP: N terminal pro-brain natriuretic peptide; 2DLVEF: ejection fraction using two dimensional echocardiography; FPRVEF and FPLVEF: first pass scintigraphy for right and left ventricular ejection fraction; V_T_/TLC: tidal volume and total lung capacity ratio; D_L_CO: diffusing capacity of lung for carbon monoxide; RV/TLC: residual volume and TLC ratio. ns: not significant. Using colors made it simple to identify the pathophysiology to which the blocks or categories (underlined) it belonged. Cells marked in yellow color indicate the relationships were significant; orange color indicates marginally significant relationship. For brevity and clarity, (r)s were not shown if they were insignificant. ^ Note: Only in the COPD group but not in the other groups, V_Tpeak_/TLC was marginally related to FPRVEF and FPLVEF (r = 0.33 and 0.30, *p* ≤ 0.1, respectively).

**Table 4 jpm-12-00703-t004:** Comparisons with the variables of interest across increasing (I), plateau (P), and decreasing (D) types of O_2_PCP in the chronic heart failure (CHF) and chronic obstructive pulmonary disease (COPD) groups (selected presentation for brevity).

	Type	I		P		D	
CHF							
		Mean	SD	Mean	SD	Mean	SD
*n*		9		13		4	
NT-proBNP		184.7	170.7	346.3	261.2	**629.8** *	410.6
CAT score		2.2	2.4	3.4	3.3	**7.5** *	4.8
@peak exercise							
V’O_2_/kg, mL/min/kg		23.8	5.6	19.4	4.6	**16.8** *	2.9
O_2_P%predicted max,%		**101.2**	12.0	80.0	19.8	85.4 *	14.7
O_2_P%pred max < 80%, *n* ^		**1**		7		1	
V’_E_/V’O_2_		**31.0**	5.2	41.3	6.7	37.6 **	2.0
FPRVEF, %		51.0	7.9	50.4	8.9	58.3	4.7
FPLVEF, %		44.2	7.9	40.7	13.2	42.4	16.1
**COPD**							
*n*		14		12		4	
OCD score		6.3	1.7	7.4	0.8	**8.9** **	1.4
mMRC score		1.1	0.7	0.6	0.5	**0.0** **	0.0
NYHAFc score		2.1	0.8	1.7	0.5	**1.0** *	0.0
@peak exercise							
FPRVEF, %		62.1	5.6	69.3	17.5	54.7	11.4
FPLVEF, %		63.5	9.4	63.5	13.3	67.8	10.5

NT-proBNP: N terminal pro-brain natriuretic peptide; CAT: COPD assessment test; V’O_2_: oxygen uptake; V’_E_/V’O_2_: minute ventilation and oxygen uptake ratio; FPRVEF and FPLVEF: first pass right and left ventricular ejection fraction; OCD: oxygen-cost diagram; mMRC: modified medical research council; NYHAfc: New York Heart Association functional classification; I: increasing; P: plateau; D: decreasing; LVEF: left ventricular ejection fraction. As no differences in all variables used here across the three types of O_2_P curve pattern in the normal controls were noted, the data were not shown for brevity. ^ Fisher’s exact test was performed for the relationship between O_2_PCP types and O_2_P%pred max < 80% or ≥80% and the result was not significant (*p* = 0.12). ANOVA: * *p* < 0.05 ** *p* < 0.01.

## Data Availability

The raw data supporting the conclusions of this article will be made available by the authors, without undue reservation.

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
