# Peer review of "The Impact of Oxygen Pulse and Its Curve Patterns on Male Patients with Heart Failure, Chronic Obstructive Pulmonary Disease, and Healthy Controls—Ejection Fractions, Related Factors and Outcomes"

_jpm, 2022, doi:10.3390/jpm12050703_

Round 1

Reviewer 1 Report

The Impact of Oxygen Pulse and its Curve Patterns on Male Patients with Heart Failure, Chronic Obstructive Pulmonary Disease, and Healthy Controls – ejection fractions, related factors and outcomes

What were the exclusion criteria for subjects used in this study?

What is the basis of using statistical methods used to analyze the data?

 Why the peak O2P% was used rather than the absolute peak exercise O2P?

Study limitations should be summarized. 

The manuscript should be checked for abbreviations errors.

Reviewer 2 Report

The authors studied the association between O2P% predicted and O2PCP and EF in patients with heart failure and COPD.

Overall, the paper is well written. I have a few minor comments for improvement.

1) Abbreviations/acronyms: The paper is filled with too many acronyms. In fact these acronyms are very distractive for the readers. You can eliminate acronization of some phrases.

If a phrase appears only once or twice, there is no need to use an acronym. For example, Oscillatory breathing (OB), cardiopulmonary exercise tests (CPETs), etc., appear only once and ROC appears only 2 times in the entire paper. Then why to acronize these terms. Just remove the acronym and say "Oscillatory breathing appears........." 

Please check all the abbreviations that appear only once.

Also please deacronize some phrases to reduce the acronyms. For example, EF can be simply written as an ejection fraction.

2) Methods: Please explain the CRP and lipids estimation in a bit more detail.

3) Figures are mislabeled. Figure 5 is before figure 4.

4) Darken Figures 3, 4, and 5 like Figure 2.

5) Decimals-rounding off: For the data with 1 to 2 digit numbers, use 1 decimal. For 3 or more digit numbers, no need to use decimals at all.

6) Tables are stand-alone and self-sustaining. That means a reader should understand the data presented in the table without referring to the text of the manuscript. So, please add more footnotes at the bottom of the table with appropriate superscripts embedded in the text of the table. Also, the table footnote should contain a type of statistical test used, abbreviations used, whether the data were mean ± SD or SE, and the significance level.

7) Figures are also stand alone. At the bottom of the figure, after the brief title, describe the data presented in the figure in a few sentences. This should also include data presented (mean±SD or mean±SE), sample size, statistical tests used, or significance level.

8) All abbreviations used in the table should be expanded in the footnote o each table.

9) All abbreviations used in the figure should be expanded in the caption (legend) of the figure. 

Reviewer 3 Report

The authors investigated several risk factors including resting and dynamic ventricular ejection fractions (EFs) and outcomes of O2P% and O2PCP types across normal subjects and patients with COPD and heart failure with reduced or mildly reduced EF (HFrEF/HFmrEF). The authors found by evaluating O2PCP and O2P% during a ramp-pattern exercise test with gas exchange and FPRV and SPECT, different relationships between ejection fraction and O2PCPs and O2P% in healthy subjects and those with COPD and 461 HFrEF/HFmrEF, and the prognosis of patients with HFrEF/HFmrEF. 

The study is very interesting for scientific community and it is also very well written and I support it for publication in the current format.

Author Response

Thank you so much.